# Pruning- and Quantization-Based Compression Algorithm for Number of Mixed Signals Identification Network

Weiguo Shen [1,2,*], Wei Wang [1], Jiawei Zhu [1], Huaji Zhou [1] and Shunling Wang [1]

1 Science and Technology on Communication Information Security Control Laboratory, Jiaxing 314033, China
2 School of Information Science and Technology, Fudan University, Shanghai 200438, China
* Correspondence: beyondswg@163.com

**Abstract:** Source number estimation plays an important role in successful blind signal separation. At present, the application of machine learning allows the processing of signals without the time-consuming and complex work of manual feature extraction. However, the convolutional neural network (CNN) for processing complex signals has some problems, such as incomplete feature extraction and high resource consumption. In this paper, a lightweight source number estimation network (LSNEN), which can achieve a robust estimation of the number of mixed complex signals at low SNR (signal-to-noise ratio), is studied. Compared with other estimation methods, which require manual feature extraction, our network can realize the extraction of the depth feature of the original signal data. The convolutional neural network realizes complex mapping of modulated signals through the cascade of multiple three-dimensional convolutional modules. By using a three-dimensional convolution module, the mapping of complex signal convolution is realized. In order to deploy the network in the mobile terminal with limited resources, we further propose a compression method for the network. Firstly, the sparse structure network is obtained by the weight pruning method to accelerate the speed of network reasoning. Then, the weights and activation values of the network are quantified at a fixed point with the method of parameter quantization. Finally, a lightweight network for source number estimation was obtained, which was compressed from 12.92 MB to 3.78 MB with a compression rate of 70.74%, while achieving an accuracy of 94.4%. Compared with other estimation methods, the lightweight source number estimation network method proposed in this paper has higher accuracy, less model space occupation, and can realize the deployment of the mobile terminal.

**Keywords:** sources number estimated; deep learning; weight pruning; parameter quantization

## 1. Introduction

With the rapid development of communication technology coupled with the influence of complex electromagnetic environment, there are more and more mixed signals in the communication system, such as the mixed signals of neighboring stars in satellite communication, the mixed signals of the same frequency caused by the increasing tension of spectrum resources. These mixed signals seriously degrade the reception performance of the communication system. The blind source separation (BBS) technology aims to separate source signals by mixed signals only when the known conditions of source signals and mixed channels are insufficient. The technology is widely used in wireless communication, an image signal, and biomedical signal. Therefore, the estimation of the number of information sources has also been a hot and difficult research area.

Early source number estimation algorithms mainly estimated the number of sources based on hypothesis testing methods. Akaike and Rissanen et al. proposed the AIC (Akaike Information Criteria) criterion and MDL (Minimum Description Length) [1,2]. Both the AIC and MDL criteria were derived in a white noise signal model and achieved better performance only under white noise conditions. However, in practice, the received signal

contains more color noise. To address this problem, H.T. Wu et al. proposed a method for estimating the number of sources based on the Geschgorin Disk Estimator (GDE) [3]. Q.T. Zhang used two uniform line arrays with differences in distance and angle to achieve the estimation of the number of sources in the background of color noise by leveraging an information–theoretic criterion approach [4]. P. Stoica proposed two source number estimation methods for two color-noise models, respectively, in which the noise covariance matrix is assumed to be a block diagonal or band diagonal array and an arbitrary matrix [5]. Hu Jun [6] proposed a source number estimation method based on the eigenspace, which reduced the number of operations of the traditional algorithm and improved the accuracy of source number estimation. Ma [7] used convolutional neural networks to design a model with three convolutional, pooling, and BN layers (3CPB). The source number estimation of single-channel mixing is achieved by processing the time-frequency image of the signal.

When there is rapid improvement in source number estimation techniques, the current algorithms have better estimation performance in the environment with a high signal-to-noise ratio but perform poorly in the case of a low signal-to-noise ratio. Since computer performance enhancement, deep learning techniques are widely applied in the field of signal processing [8–11]. In particular, CNN can replace the tedious manual extraction of data features by supervised learning with nonlinear fitting capabilities and thus achieve tasks such as recognition and classification. To further improve the feature extraction performance of CNN, scholars have continuously tried to increase the depth and width of the network. However, as the network model deepens, the network suffers from problems such as gradient dispersion and gradient explosion, which affect the training of the network. In 2015, the inception module was proposed to widen the network model in the GoogleNet network [12]. In the same year, the residual module was proposed in the ResNet network to solve the problem of continued training of its 152-layer network [13]. In [14,15], CNN increased with the number of layers and width of the network, although the accuracy rate increased, at the same time, the computational resources required by the network increased sharply, and training becomes more difficult. This is one of the main reasons why deep convolutional neural networks are difficult to be applied in mobile applications with limited computational resources and high real-time requirements. To solve the above problems, researchers have proposed many theories to achieve compression of large models, such as reducing the network model size, decreasing the number of parameters in the network model [16–19], and reducing the number of network model parameter bits [19–21]. Low-power mixed-signal design technology in semi-programmable ASICs has been studied in the literature [22]. To balance the energy and precision of neural networks, Song Han et al. applied Huffman coding with pruning and quantization techniques, reducing the storage space by 35 times [23]. In the case of the ResNet network, Zhao, M et al. achieved network compression through a combination of knowledge distillation and pruning, resulting in faster reasoning speed and higher accuracy than the original network [24]. Scholars have classified pruning and quantization into various forms, such as static pruning, dynamic pruning, element-wise, channel-wise, shape-wise, filter-wise, layer-wise, and even network-wise pruning [25]. Additionally, some researchers [26] have explored network architecture search algorithms to create optimized network models.

In this paper, complex data is reconstructed according to the characteristics of complex modulated signals in wireless communication. Additionally, a convolutional neural network which can extract the depth features of signals is designed. A robust estimation of the number of sources under low SNR is achieved. Aiming at complex network models, the weight pruning and parameter quantization methods are combined to improve the model reasoning speed and model compression. Thus, it can be deployed on the mobile terminal with limited resources.

The main contributions of this work are summarized as follows:

1.  In this paper, we studied a source number estimation network (SNEN) which can realize robust number estimation;

2. Redundant connections in convolutional neural networks are removed by weight pruning. The sparse network structure is obtained, and network reasoning speed is accelerated;

3. The weights and activation values of convolutional neural networks are quantified. The model is quantized to different bits and compared to obtain the optimal quantized bits;

4. A combination of weight pruning and parameter quantization compression method is performed on the network to achieve a lightweight source number estimation network.

The remaining parts of this paper are organized as follows. Section 2 introduces the system model and analyzes the modeling of mixed multi-dimensional independent statistical source signals. Section 3 focuses on the reconstruction of complex signals, the construction of neural networks, and methods for network lightweight. Section 4 analyzes the experimental results. Finally, we present the research conclusions in Section 5.

## 2. System Model

In this paper, the transmitting source uses $M$ dimensional linear instantaneous mixed signal, and the receiving end adopts single-channel observation. In practice, linear instantaneous mixing model is the most common type of source signal mixing.

For ease of reading, Table 1 is a summary table of symbols mentioned in Section 2.

**Table 1.** Symbols and their brief description/definition in Section 2.

| Symbol | Description/Definition |
|:------:|:----------------------:|
| $a_{ij}$ | unknown mixing coefficient |
| $s_i(t)$ | source signal |
| $x_j(t)$ | mixed signal |
| $\boldsymbol{n}(t)$ | $N \times 1$ dimension noise vector |
| $\boldsymbol{x}(t)$ | $N \times M$ dimension mixed matrix |

It is assumed that a source signal with $M$ dimension and mutual statistical independence passes through the transmission channel, and $N$ dimension observation signal can be observed by receiving sensor. The mathematical modeling of observation signal can be shown as follows:

$$x_j(t) = \sum_{i=1}^{M} a_{ij} s_i(t) \tag{1}$$

where $a_{ij}$ is an unknown mixing coefficient, $s_i(t)$ is a source signal, and $x_j(t)$ is a mixed signal, $i \in [1, 2, \cdots, M]$, $j \in [1, 2, \cdots, N]$. Expand Formula (1):

$$\begin{bmatrix} x_1(t) \\ x_2(t) \\ \vdots \\ x_N(t) \end{bmatrix} = \begin{bmatrix} a_{11} & a_{12} & \cdots & a_{1M} \\ a_{21} & a_{22} & \cdots & a_{2M} \\ \vdots & \vdots & \ddots & \vdots \\ a_{N1} & a_{N2} & \cdots & a_{NM} \end{bmatrix} \begin{bmatrix} s_1(t) \\ s_2(t) \\ \vdots \\ s_M(t) \end{bmatrix} \tag{2}$$

The above formula can be expressed as follows:

$$\boldsymbol{x}(t) = \boldsymbol{A}\boldsymbol{s}(t) \tag{3}$$

In the actual transmission process, noise will interfere, and then the mixed signal is

$$\boldsymbol{x}(t) = \boldsymbol{A}\boldsymbol{s}(t) + \boldsymbol{n}(t) \tag{4}$$

where $\boldsymbol{n}(t)$ represents a $N \times 1$ dimension noise vector, $s(t)$ represents $M \times 1$ dimension vector source signal, and $x(t)$ is a $N \times M$ dimension mixed matrix.

Under the condition of single channel, the dimensions of the observation signals for $N \times 1$, $\boldsymbol{A} = [a_1, a_2, \cdots, a_M]$ become mixed coefficient vector. Source number estimation

problem is based on the observed signal $x(t)$ estimate $M$, which can be expressed as $M$ classification problem of a class.

## 3. Lightweight Source Number Estimation Network Method

Most of the current convolutional neural networks are implemented for real numbers as objects, so the fitting of complex mappings cannot be realized. Additionally, convolution operation will break the structural relationship between real and imaginary parts of a complex signal. However, if the convolution layer is redefined, complex convolution will make the operation more complicated and the realization of the network more difficult. In order to solve the above problems, the mixed signal number identification network framework is deployed on the mobile terminal. A source number estimation network (SNEN) is constructed in this paper. Furthermore, we carried on the network weight pruning and parameter quantization operation and reduced the network overhead. The workflow is shown in Figure 1.

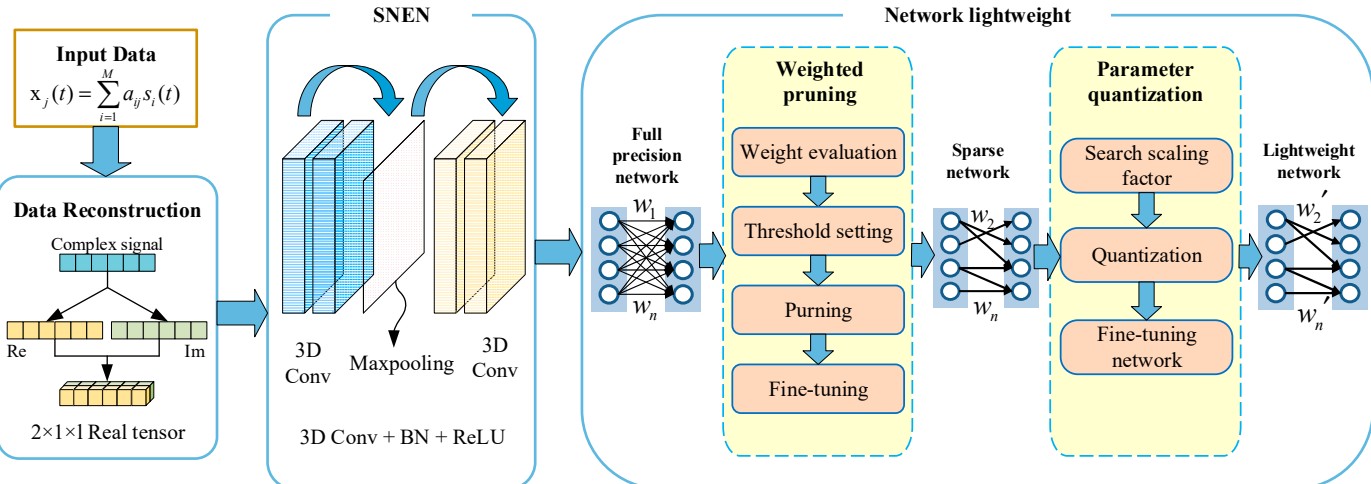

**Figure 1.** Lightweight source number estimation network flow.

Firstly, the input mixed signal is reconstructed into a three-dimensional real tensor with length, width, and number of channels. The three-dimensional real tensor is input into the source number estimation network for training. Then, weight pruning is carried out on the network after convergence to remove the connections with less contribution. Finally, the network is greatly compressed by quantifying the weight.

For ease of reading, Table 2 is a summary table of symbols mentioned in Section 3.

**Table 2.** Symbols and their brief description/definition in Section 3.

| Symbol | Description/Definition |
|---|---|
| $x(n)$ | input data |
| $w(n)$ | convolution kernel weights |
| $\mathrm{Re}\{\cdot\}$ | real part |
| $\mathrm{Im}\{\cdot\}$ | imaginary part |
| $\theta$ | the set of all trainable parameters |
| $P(\cdot)$ | posterior probability |
| $y^{(m)}$ | observed data |
| $W_k$ | connections between neurons in the $k$ layer of the neural network |
| $L$ | the number of layers of the neural network |
| $A(\cdot)$ | the accuracy of the neural network |
| $\odot$ | he Hadamard product |

**Table 2.** *Cont.*

| Symbol | Description/Definition |
|---|---|
| $W_k^i$ | the *i-th* weight of the *k-th* network layer |
| $\varsigma_k$ | all neurons of the *k-th* network layer |
| $n_0$ | threshold interval |
| $V_n$ | the *n-th* threshold |
| $\lambda$ | regular coefficient |
| $X$ | the parameter to be quantified |
| $\lfloor \cdot \rfloor$ | integer operation |
| $S_X$ | the scaling factor of $X$ |

### 3.1. Data Preprocessing

In this section, the process of complex convolution is studied, and the real and imaginary parts of complex signals are disassembled and re-spliced. The complex signal is reconstructed into a three-dimensional real tensor with length, width, and number of channels. The reconstructed real tensors are beneficial to the operation of convolution and reduce the difficulty of network implementation.

Digital modulated signals are mostly in complex form, which requires the convolutional neural network to realize convolution operation in complex domains. Compared with real convolution, complex convolution is more complex, which can be expressed as

$$
\begin{aligned}
\hat{s}(n) \ &= w(n) * x(n) \\
&= [\mathrm{Re}\{w(n)\}+j\mathrm{Im}\{w(n)\}] * [\mathrm{Re}\{x(n)\}+j\mathrm{Im}\{x(n)\}] \\
&= [\mathrm{Re}\{w(n)\} * \mathrm{Re}\{x(n)\} - \mathrm{Im}\{w(n)\} * \mathrm{Im}\{x(n)\}]+ \\
&\quad j[\mathrm{Re}\{w(n)\} * \mathrm{Im}\{x(n)\} + \mathrm{Im}\{w(n)\} * \mathrm{Re}\{x(n)\}]
\end{aligned}
\tag{5}
$$

where $x(n)$ denotes the input data, and $w(n)$ denotes the convolution kernel weights. Its matrix form is as follows:

$$
\begin{bmatrix} \mathrm{Re}\{\hat{s}(n)\} \\ \mathrm{Im}\{\hat{s}(n)\} \end{bmatrix} = \begin{bmatrix} \mathrm{Re}\{w(n)\} & -\mathrm{Im}\{w(n)\} \\ \mathrm{Im}\{w(n)\} & \mathrm{Re}\{w(n)\} \end{bmatrix} * \begin{bmatrix} \mathrm{Re}\{x(n)\} \\ \mathrm{Im}\{x(n)\} \end{bmatrix}
\tag{6}
$$

where $\mathrm{Re}\{\cdot\}$ and $\mathrm{Im}\{\cdot\}$ denote the real and imaginary parts of the complex numbers, respectively. Both the real and imaginary parts of the complex convolution operation structure in Formulas (5) and (6) are linear combinations of the real and imaginary parts of the input data after convolution with each other.

As shown in Figure 2, the real and imaginary parts of the complex sequence of $1 \times l$ are separated, the imaginary part sequence is placed behind the real sequence, and the complex sequence is reconstructed into a 3-dimensional real tensor with width, length, and number of channels of size $2 \times 1 \times l$.

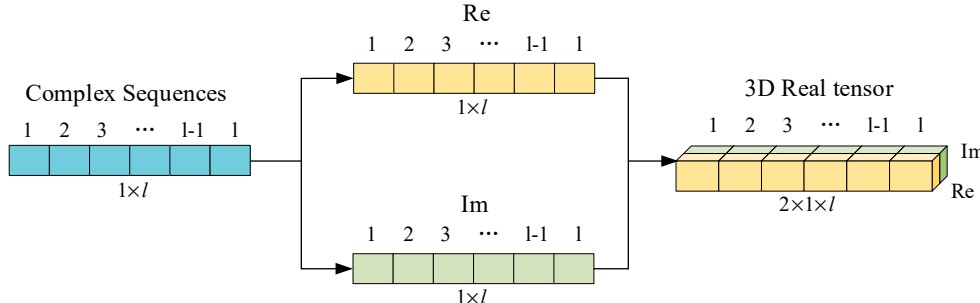

**Figure 2.** Data reconfiguration diagram.

### 3.2. Source Number Estimation Network

In this section, the source number estimation network is constructed to process the reconstructed 3D real tensor. Compared with other current solutions, this architecture is

simple to implement. It can effectively realize the learning of complex mapping. Additionally, it can avoid the problem that the activation function and optimization algorithm cannot obtain theoretical support in the complex domain.

The specific implementation of mixed-signal feature extraction is as follows:

1.  Perform data reconstruction on the input complex data $x(n)$, and reconstruct the data into a three-dimensional tensor;
2.  Input the reconstructed data into each of the two convolutional layers, the two convolutional operations corresponding to the real part output and the imaginary part output in Formula (5);
3.  Reconstruct the 2-output data in 2 into a complex sequence. Then, input sequence into the next layer.

In Formula (5), the complex convolution process can be divided into two parts: the real part $\mathrm{Re}\{w(n)\} * \mathrm{Re}\{x(n)\} - \mathrm{Im}\{w(n)\} * \mathrm{Im}\{x(n)\}$ and the imaginary part $\mathrm{Re}\{w(n)\} * \mathrm{Im}\{x(n)\} + \mathrm{Im}\{w(n)\} * \mathrm{Re}\{x(n)\}$. In order to realize the mapping of complex signals, two convolutional modules with the same structure are designed in Figure 3, respectively, corresponding to the real and imaginary parts of the output in Formula (5). Then, the output data is reconstructed to realize the mapping of complex convolution.

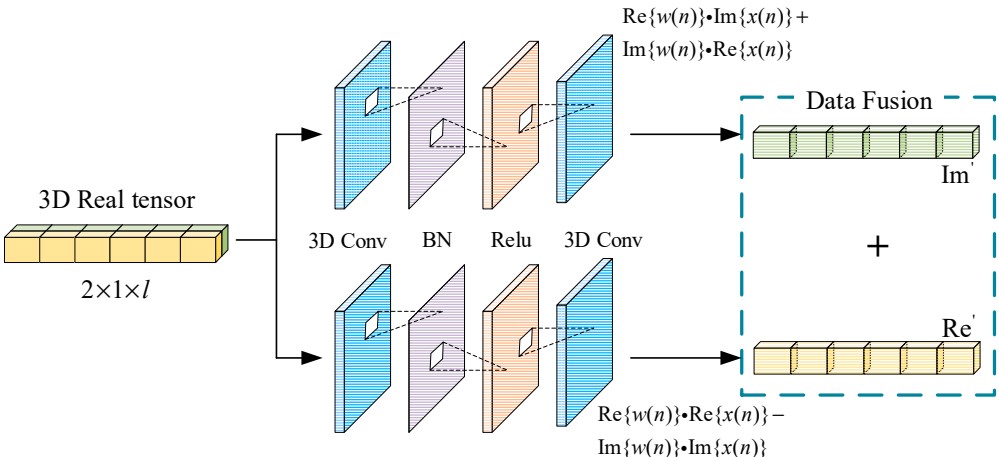

**Figure 3.** Feature extraction module.

Each convolution operation shown in Figure 3 consists of a 3D real convolution layer, Batch Normalization layer (BN) and activation function layer. The first 3D convolutional layer uses a convolutional kernel of size $2 \times 1 \times m$; the second 3D convolutional layer uses a size $2 \times 1 \times 1$ in order to reduce the dimensionality of the output data while converting the 3-dimensional data into the real and imaginary parts of the output data in Formula (5).

The activation function uses ReLU. This function is linear, which means that it has desirable properties of linear activation functions when using backpropagation to train neural networks. Additionally, it can avoid the problem of disappearing gradient. It is simpler in calculation and easy to optimize.

Using the above CNN architecture, a network that can realize source number estimation is designed. Figure 4 shows the structure diagram of source number estimation. The network contains four convolution units, which contain a BN layer to control gradient explosion and overfitting. After the convolution unit, there is a pooling layer to reduce the output feature size. Finally, feature mapping is realized by a fully connected layer. The dropout layer is added to the full-connection layer to prevent over-fitting and improve model generalization ability. The mixed signals are first extracted by CNN, then feature mapped by a fully connected layer, and finally, the classification results are output by the Softmax layer.

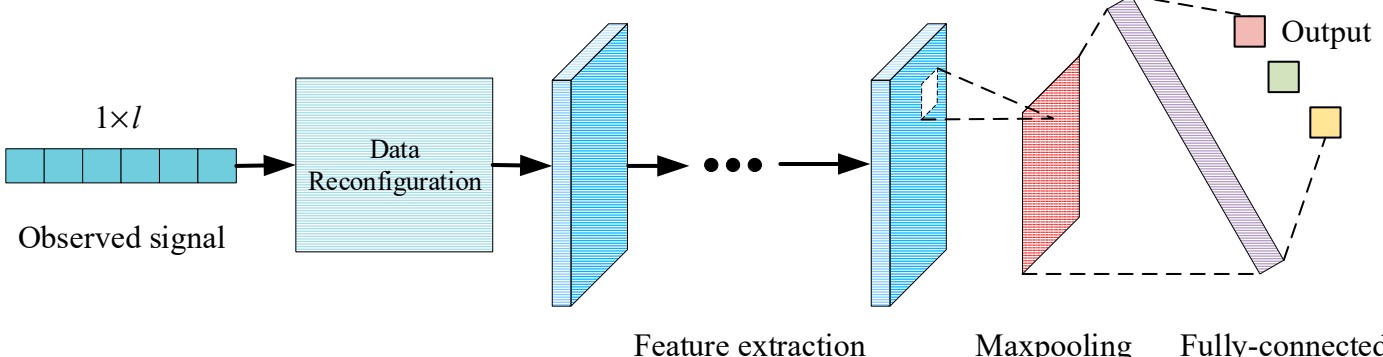

**Figure 4.** Number estimation network structure model.

The 3D convolutional module extracts the depth features of the mixed signals and inputs them to the softmax classifier for decision output. Through the softmax function, the probability of each category can be obtained. Finally, the category with the largest probability value is determined to be the final output.

The training objective of the classifier can be expressed by the maximum likelihood function:

$$L(\theta) = P(T \mid Y; \theta) = \prod_{m=1}^{M} \prod_{i=0}^{I} \left( P_{\theta|H_i} \left( y^{(m)} \right) \right)^{g^{(m)}(t_i)} \tag{7}$$

where $\theta$ is the set of all trainable parameters. $P(\cdot)$ is the posterior probability derived from the training parameter $\theta$. $y^{(m)}$ is the observed data.

Take the logarithm of the above equation:

$$
\begin{aligned}
l(\theta) \quad &= \log L(\theta) \\
&= \sum_{m=1}^{M} \sum_{i=0}^{I} g^{(m)}(t_i) \log P_{\theta|H_i} \left( y^{(m)} \right)
\end{aligned}
\tag{8}
$$

Therefore, the cost function is

$$
\begin{aligned}
J(\theta) \quad &= -\frac{1}{M} l(\theta) \\
&= -\frac{1}{M} \sum_{m=1}^{M} \sum_{i=0}^{I} g^{(m)}(t_i) \log P_{\theta|H_i} \left( y^{(m)} \right)
\end{aligned}
\tag{9}
$$

The gradient descent method is used to make the cost function decrease continuously and update $\theta$ value to make it converge.

$$\theta^* = \arg\max_{\theta} P(T|Y; \theta) \tag{10}$$

After the value of $\theta$ converges, the maximum likelihood function of the classifier is obtained, and the output probability is finally obtained.

Compared with the traditional CNN, the source number estimation network used in this paper can realize the learning of complex mapping. The network avoids the problem that the activation function and optimization algorithm cannot obtain theoretical support in complex domains. Additionally, it successfully avoids complex signal feature extraction and complex network implementation difficulties. The source number estimation network achieves the number estimation of mixed multi-signal by the stacking of multiple convolutional modules and traditional fully connected layers. The stacking of the multi-layer network structure makes the model structure extremely deep, and the amount of mapping relationship data between the nodes stored in the model is extremely large. A large amount of node mapping data makes the recognition accuracy of the network effectively improved. However, it is not helpful for our deployment on the mobile side, where resources are limited. In order to reduce the complexity of the network and ensure a high accuracy rate.

In this paper, we will continue to use a combination of weight pruning and parameter quantization to achieve compression of the convolutional neural network to meet the needs of deployment in mobile.

### 3.3. Threshold-Based Weight Pruning Algorithm

Aiming at the problems of redundant parameters, complex models and slow reasoning speed, the weight pruning method is used in this paper to reduce the structural complexity of the source number estimation network. The changes in neuron weight before and after pruning are shown in Figure 5. Firstly, regularization is added to the target detection network model for sparse training to obtain the scaling factor. Then, the proportion of channel importance in the convolution layer is calculated according to the distribution of the scaling factor. Finally, the convolution layer with a small contribution to the network model is cut off by calculating the dynamic threshold based on the scaling factor.

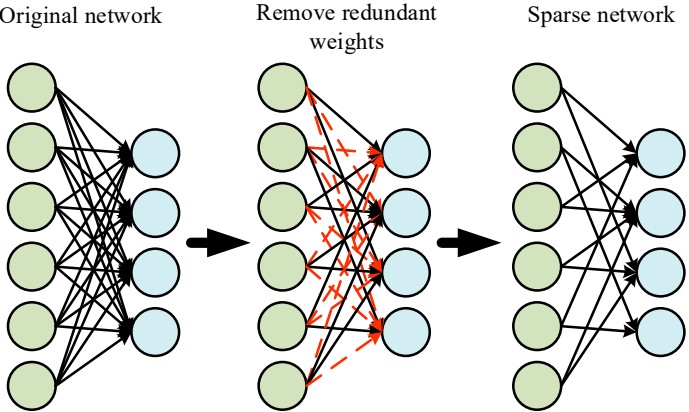

**Figure 5.** Neuronal changes before and after pruning.

Suppose the model parameters are expressed in terms of $W$, $W = \{W_1, W_2, \cdots, W_L\}$, where $W_k$ represents the connections between neurons in the $k$ layer of the neural network, and $L$ represents the number of layers of the neural network. The clipping of neurons is represented by adding a mask to each neuron. The parameter masks of neurons in the model are represented by $T$, $T = \{T_1, T_2, \cdots, T_L\}$, and the value of each mask is 0 or 1. If the neuron is cut off, its corresponding mask value will be set to 0; otherwise, it will be set to 1. Therefore, the target of pruning can be set as

$$\min \sum \|T_k\|_0, \ s.t. |A(W \odot T) - A(W)| < \theta, \tag{11}$$

where $A(\cdot)$ represents the accuracy of the neural network, $\odot$ represents the Hadamard product, and $W$ represents the unpruned parameters in the neural network. $W \odot T$ specifies the parameter after the mask is added.

Equation (13) is to minimize the number of non-zero parameters in the model when the accuracy of the model does not exceed the given threshold value $\theta$. Before the model is pruned, the mask of all weights will be set to 1, indicating that all weights are not pruned. When pruning, the pruning threshold is calculated by the algorithm. When the absolute value of the weight is less than the threshold, the corresponding mask value will be set to 0, which means that the weight can be cut off. Finally, by using the Hadamard product, the parameter value corresponding to the mask value 0 is calculated as 0, thus realizing the purpose of parameter pruning. The mask value can be obtained by Formula (14):

$$T_k = p(W_k^i), \forall i \in \xi_k, \tag{12}$$

where $W_k^i$ represents the *i-th* weight of the *k-th* network layer, and $\xi_k$ represents all neurons of the *k-th* network layer. In addition, $p(\cdot)$ is the pruning function, which is used to determine whether the neuron is pruned.

In order to obtain the pruning threshold, firstly, the weight $W$ of all parameters in the neural network is obtained. The maximum absolute value $W_{max}$ and the minimum absolute value $W_{min}$ of parameter weights are calculated. Then, the threshold interval size $N$ is set. Additionally, the equal interval is divided between the maximum value and the minimum value of the absolute value of the parameter weight to obtain the threshold interval value $n_0$.

$$n_0 = \frac{W_{max} - W_{min}}{N} \tag{13}$$

With the threshold interval value, you can set a threshold between the maximum and minimum of the absolute value of the weight. The *n-th* threshold can be expressed as

$$V_n = W_{min} + n_0 \cdot n, \forall n \in [1, N]. \tag{14}$$

For each test threshold, the accuracy of the model under the current test threshold is tested. If the accuracy of the model falls within the given threshold, the next threshold will be tested; otherwise, the pruning will be stopped. Finally, the optimal threshold $V_{threshold}$ is obtained:

$$V_{threshold} = \underset{V_n}{argmin} \sum \|T_k\|_0. \tag{15}$$

In the stage of network searching for optimal threshold. Adding regularization in the process of network training can generate sparse weights, which can be used to train large and dense networks. That is, the weight of the high contribution to the network should be enlarged, and the weight of the low contribution to the network should be reduced. This method can accelerate the reasoning speed of the network. When L2 Regularization is added, the Cross-Entropy Loss function is:

$$C = -\frac{1}{n} \sum_x [ylna + (1 - y)ln(1 - a)] + \frac{\lambda}{2n} \sum_W W^2, \tag{16}$$

where $W$ is the model parameter, and $\lambda$ is the regular coefficient.

Weighted Pruning

In the network training, the weight values of each node are divided into positive and negative. The negative weights with large absolute values have a very important influence on the network, so pruning cannot be done directly according to the size of the weight values. In addition, the distribution of weights in each layer of the source number estimation network is different, so the pruning of the network is carried out in layers. Therefore, the threshold $t$ for pruning in this paper is chosen as the standard deviation of the weights of each layer, and pruning is performed according to the absolute value of the weights. The specific steps of the pruning algorithm are as follows:

1.  Train the source number estimation network normally until the network converges so that the network is fully learned;
2.  Set a threshold $t$ and prune the connections with absolute values of weights less than the threshold $t$, transforming the dense network into a sparse network;
3.  Fine-tune the resulting sparse network to reduce the loss of accuracy due to pruning.

### 3.4. Parameter Quantization

In order to compress the size of the source number estimation network model, the parameter quantization method is used to reduce its space occupation. Parameter quantization works by converting the weight and activation values of 32-bit floating-point numbers stored by the model into fixed-point numbers with lower bits. This operation can reduce the bit width of parameters, reduce memory consumption, and achieve model compression. At the same time, the model compression method combining parameter quantization and weight pruning not only increases the model reasoning speed but also reduces the space

occupation of the model. The method achieved a better compression effect. The process of parameter quantization is shown in Figure 6.

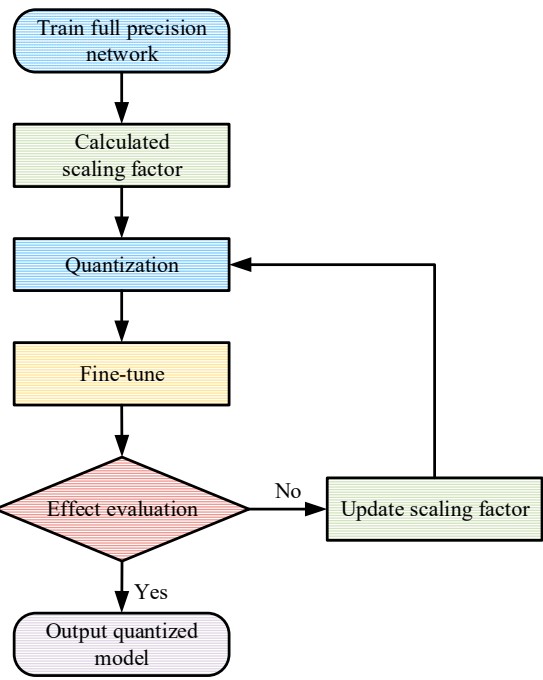

**Figure 6.** Parametric quantization flow chart.

The specific steps of parameter quantization are as follows:

1. The full precision source number estimation network is trained well, and the scaling coefficients of weight and activation values are calculated according to the algorithm;
2. Weights and activation values are quantized according to the scaling factor. Each weight and activation value is multiplied by the scaling factor and rounded;
3. Judge the quantization effect. If the precision is not satisfied, the quantization number is selected again.

The first thing is to find the right scaling factor. For a network consisting of $N$ convolutional layers or fully connected layers, the weight and the scaling factor of the activation value of each convolutional layer need to be calculated. Therefore, it is necessary to study the common characteristics of different convolution layers in the network model and adopt the same strategy for quantization to greatly simplify the quantization process. When quantizing weight and activation values, record the maximum and minimum values of weight and activation values. The maximum range of weights is simple to obtain, with the maximum range of weights $[\min, max]$. The activation values are smoothed by using the Exponential Moving Average (EMA) function, and the activation values are obtained from the range $[\min, max]$. The EMA takes a value as close to 1 as possible. If the maximum value is directly used as the upper bound of quantization, the quantization function is

$$Q(X) = \frac{\lfloor X \times S_X \rfloor}{S_X},\qquad(17)$$

where $X$ is the parameter to be quantified in the neural network. $\lfloor \cdot \rfloor$ indicates the integer operation. $S_X$ is the scaling factor of $X$, scaling the weights and activation parameters to the appropriate range. If directly according to the weight or activation, quantify the maximum of a quantitative target bit number $n$, the maximum parameter $X$ for $x_{max} = max(|X|)$, and the distribution of $X$ zoom coefficient of $S_X$ for

$$S_X = \frac{2^{n-1}}{x_{max}}.\qquad(18)$$

After the scaling factor is obtained, the network can be quantized. However, in order to further improve the quantization effect, the network needs to be fine-tuned to improve the accuracy. That is, the optimal scaling factor $S_X$ is found. This scaling factor is generally larger than that used to quantify directly with weights or activated maximum values. At this point, the value $\lfloor X \times S_X \rfloor$ after parameter quantization may exceed the range that $n$ bits can represent $[-2^{n-1}, 2^{n-1} - 1]$, so it is necessary to clip the excess part. Truncated and then divided by the same scaling factor to restore the weight to its original range, the final quantization function is

$$Q(X) = \frac{min(max(\lfloor X \times S_X \rfloor, 1 - 2^{n-1}), 2^{n-1} - 1)}{S_X}, \tag{19}$$

where $min(max(\cdot, \cdot), \cdot)$ is a truncation operation. The truncation operation introduces the Clipping Error. However, due to the improvement of quantization resolution, Quantization Error will decrease correspondingly. Therefore, there will be a better quantization effect in the end.

Finally, the performance of the quantized network is analyzed to determine whether the compression ratio and accuracy meet the requirements. For the network with high accuracy loss, the scaling coefficient should be re-selected. Otherwise, quantization ends. Output quantized network.

In what follows, the network lightweight process Algorithm 1 of this paper is described in detail.

---

**Algorithm 1. Network pruning and quantization algorithm.**

---

**Input:** $W$, $N$, $a_0$, $\theta_a$, $X$, $S_X$;
Where $W = \{W_1, W_2, \ldots, W_L\}$ is the weight of the pre-training model; $N$ is the threshold interval; $a_0$ is the initial accuracy of the model; $\theta_a$ is the threshold of accuracy; $X$ is the parameter to be quantized; $S_X$ is the scaling factor.
**Output:** $W_k^i$, $Q(X)$;
Where $W_k^i$ is the weight after training; $Q(X)$ is the quantized parameter.

1:  **function** Train ($W$, $N$, $a_0$, $\theta_a$, $X$, $S_X$)
2:  Obtain the maximum $W_{max}$ and minimum $W_{min}$ of the absolute value of the model parameters;
3:  The threshold interval $n_0$ can be obtained from Equation (13);
4:  The test threshold $V$ can be obtained from Equation (14);
5:  **for** $V_n$ in $V$ **do**
6:  　　**for** $W_k^i$ in $W$ **do**
7:  　　　　**if** $\left| W_k^i \right| < V_n$ **then**
8:  　　　　　　$T_k = p(W_k^i)$
9:  　　　　**end if**
10: 　　**end for**
11: 　　**if** $|a - a_0| > \theta_a$ **then**
12: 　　　　**return** $W_k^i$;
13: 　　**end if**
14: **end for**
15: Quantization of weight and activation value:
　　$Q(X) = \frac{min(max(\lfloor X \times S_X \rfloor, 1 - 2^{n-1}), 2^{n-1} - 1)}{S_X}$
16: **end function**

---

## 4. Results

In order to verify and analyze the superiority of the algorithm in this paper, four sets of experiments are conducted in this paper: (1) to verify the effectiveness of the source number estimation network to achieve the estimation of the number of sources; (2) to verify the effectiveness of the threshold-based weight pruning algorithm on the proposed source number estimation network; (3) to quantize the weights and activation values of the source number estimation network to 16 bit, 8 bit, and 4 bit, respectively, and to compare the

quantized network with the unquantized floating-point model. (4) The model weights are pruned to obtain the sparsity network structure, and then the sparse network is quantized to further compress the network.

### 4.1. Performance Analysis of Source Number Estimation Network

In this paper, a source number estimation network is designed to process commonly used digitally modulated signals. The data set used to train the source number estimation network is generated by Matlab simulation software, which simulates the generation of mixed signals containing BPSK, QPSK, FSK and 16 QAM. The signal sampling rate is 48 KHz, the number of sampling points is 1024, and each symbol is sampled by 8 points. The SNR of the signals obeys the following uniform distribution: $SNR \in [-20 \, dB, 20 \, dB]$, with an interval of 5 dB. The training set contains 1,000,000 samples, and the number of samples for each type of signal at different SNRs is 100. The experimental equipment used in this study was a PC with a Windows operating system, equipped with an Intel i5-12600 CPU and an RTX 3080 GPU. The neural network was built using the TensorFlow framework. In this section, the designed source number estimation network is compared with the existing number estimation network, and the accuracy is used as the performance evaluation index.

The internal structure of the source number estimation network designed in this paper is more complex, in which the number of 3D convolutional units is 4, the size of the convolutional kernels in each branch convolutional layer is $2 \times 1 \times 8$, and the number and size of the convolutional kernels in each convolutional layer are equal.

In this section, the source number estimation network is compared with the extant number estimation methods, using accuracy as a performance evaluation metric. Two conventional source number estimation methods, AIC and MDL, are implemented using Matlab and compared with the method proposed in this paper. The method proposed in this paper is then compared with other convolutional neural networks.

Figure 7a shows the accuracy of the source number estimation network with the AIC criterion and MDL criterion for estimating the number of sources at different SNRs. It can be seen that when SNR < 5 dB, the accuracy of the traditional method and source number estimation network method increases with the increase of SNR. When SNR > 5 dB, the accuracy of the 3 methods tends to be gentle, and all of them can reach more than 96%. Compared with traditional methods, the source number estimation network performs better at low SNRs. Even when SNR = −20 dB, it still has 84% accuracy. When SNR > −10 dB, the identification accuracy can reach 95%, and the number of sources can be estimated robustly. However, the traditional method cannot realize the robust estimation of the number of sources under the low SNRs.

Figure 7b shows the accuracy of the number of sources estimated by the source number estimation network and other CNN [7] at different SNRs. Another CNN contains three convolutional layers, each of which is followed by a pooling layer and a BN layer. Other CNNs are able to achieve an estimation accuracy of over 90% at SNR > −10 dB. However, at lower SNRs, the accuracy of the network estimation drops sharply. By comparing with traditional methods and other convolutional neural networks, the method proposed in this paper has a strong anti-interference ability. The method proposed in this article achieves higher accuracy compared to other approaches by reconstructing complex signals into three-dimensional real-valued tensors and utilizing three-dimensional convolutional units to extract signal features while preserving the interdependence between the real and imaginary parts. When the SNR is less than −15 dB, the accuracy is 85%, and when the SNR is more than −10 dB, the accuracy is more than 93%. Therefore, the method proposed in this paper can also achieve effective estimation under low SNR.

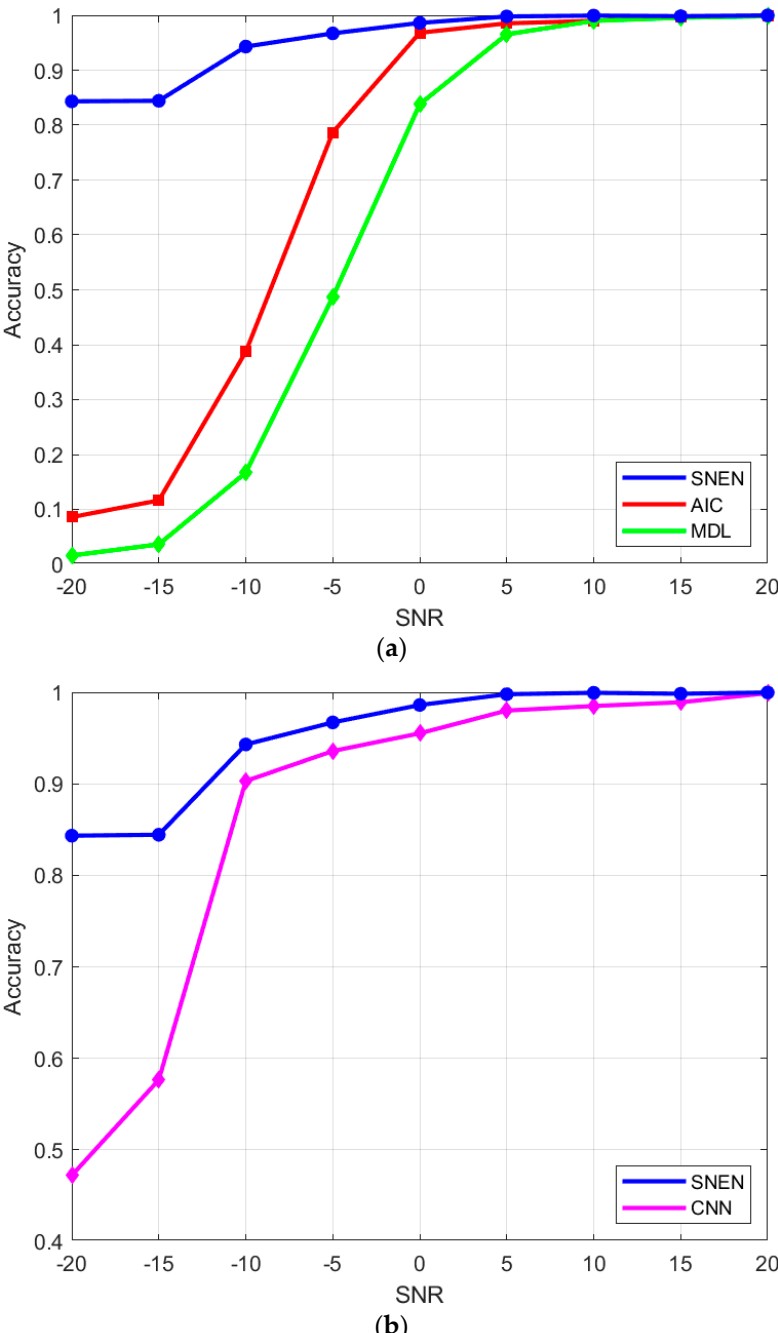

**Figure 7.** Comparison chart of various source number estimation methods. (**a**) shows the accuracy of this method compared to traditional estimation methods. (**b**) shows the accuracy of the present method compared with other CNN.

### 4.2. Performance Analysis of Network after Pruning

This section verifies the effectiveness of the threshold-based pruning algorithm on the proposed source number estimation network. The network contains several convolutional and several fully connected layers, and the structure is more complex. Table 1 lists the number of weights, pruning thresholds, and pruning rates for each layer of the proposed network in this paper.

From Table 3, it can be seen that after the network was pruned, the number of weights was reduced from $319 \times 10^4$ to approximately $53.6 \times 10^4$. The pruning rate reached 83.20%, while the accuracy of the pruned model was reduced by approximately 3.72% compared with the original source number estimation network. Moreover, it can be seen from Table 1

that the parameters of the convolutional neural network are mainly concentrated in the fully connected layer. Additionally, the cropping rate of the fully connected layer is higher than that of the convolutional layer.

**Table 3.** Weight number, Threshold, and Pruning rate of different layers in source number estimation network.

| Layer | Weights Number | Threshold | Pruning Rate |
|---|---|---|---|
| Conv1 | $1.4 \times 10^4$ | 0.07125 | 43.40% |
| Conv2 | $24 \times 10^4$ | 0.10174 | 68.42% |
| Conv3 | $46 \times 10^4$ | 0.06571 | 78.49% |
| FC1 | $200 \times 10^4$ | 0.16179 | 87.18% |
| FC2 | $48 \times 10^4$ | 0.11726 | 78.98% |
| Total | $319 \times 10^4$ | - | 83.20% |

Figure 8 shows the comparison of the accuracy of the pruned lightweight source number estimation network and the original network. It can be seen that, compared with the original network, the accuracy rate of the network after pruning is reduced by 3.72%. However, it is still better than the traditional AIC, MDL, and other convolutional neural network methods. After pruning, the recognition accuracy of the network remains above 80% under low SNR. After pruning, the network accuracy decreased within an acceptable range, and the network weight was greatly reduced. The network compression effect is obvious, so the pruning algorithm of source number estimation network is feasible.

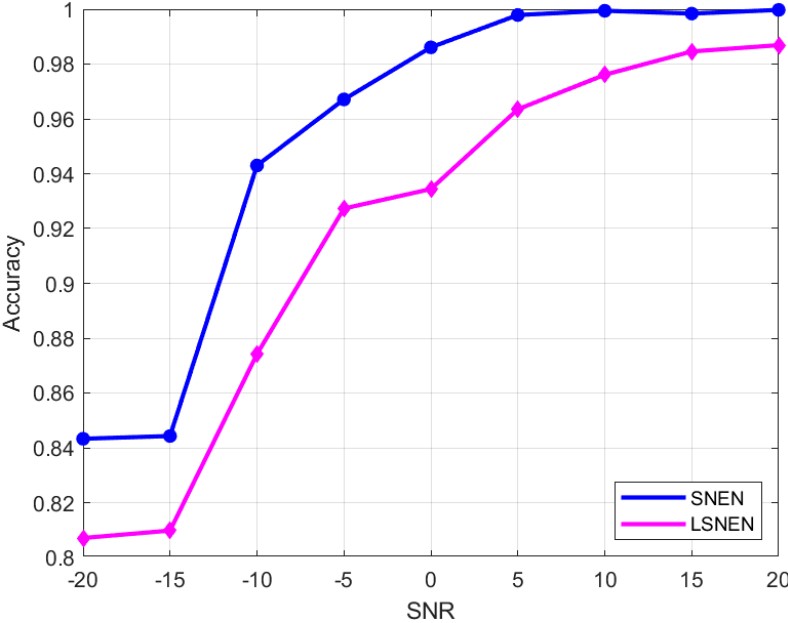

**Figure 8.** Comparison chart of accuracy before and after pruning.

*4.3. Performance Analysis of Network after Quantization*

This section uses the quantization-aware-training method to quantize the weights and activation values of the source number estimation network, compares the changes in the compression ratio and accuracy of the model when quantizing it to 16 bit, 8 bit, and 4 bit, and selects the most suitable number of quantization bits.

Table 4 presents the results of the source number estimation network before and after quantization. Where $W/A$ denotes the number of bits quantized by the weights and activation values of the network structure, respectively. The parameter $\tau$ is the ratio of the change in model size to the change in accuracy. The larger value of $\tau$ indicates that the number of quantized bits is optimal at this time. The experimental results show that when the network is compressed to 16 bits, the size of the source number estimation network

model is reduced from 12.92 MB to 6.39 MB, the compression ratio reaches 50.54%, and the average accuracy rate can reach 96%, which is 99.28% of the original network; when the network is compressed to 8 bits, the size of the source number estimation network model is reduced from 12.92 MB to 4.16 MB, the compression ratio reaches 67.80%, and the average accuracy rate can reach 95.8%, which is 99.07% of the original. When the network is compressed to 4 bits, the size of the source number estimation network model is reduced from 12.92 MB to 3.38 MB, the compression ratio reaches 73.83%, and the average accuracy rate can reach 91.3%, which is 94.42% of the original network. Although the 16-bit quantization cannot compress the network significantly, its average accuracy rate decreases less. Additionally, although the quantization to 4 bits can make the network model compressed to a small size, the accuracy of the network is greatly decreased, resulting in a smaller $\tau$ compared with others. The quantization of the parameters of the source number estimation network is of great importance for its deployment on mobile, where resources are limited.

**Table 4.** Source number estimation network quantification results.

| Bits (W/A) | Model Size/MB | Average Accuracy | Compression Ratio | Declining Accuracy | $\tau$ |
|---|---|---|---|---|---|
| 32/32 | 12.92 | 96.7% | - | - | - |
| 16/16 | 6.39 | 96.0% | 50.54% | 0.007 | 72.20 |
| 8/8 | 4.16 | 95.8% | 67.80% | 0.009 | 75.33 |
| 4/4 | 3.38 | 91.3% | 73.83% | 0.054 | 13.67 |

*4.4. The Combination of Pruning and Quantification*

Analysis of the above experimental data shows that quantization of the source number estimation network model to 8 bit ensures high accuracy and the best network compression. In this section, the pruning algorithm is combined with the quantization algorithm to obtain a sparse network structure, and the network model weights and activation values are quantized to 8 bits.

Table 5 shows the results of the weight pruning and quantization operations on the source number estimation network. The model compression ratio reaches 70.74%, and the accuracy rate is 94.4%. The accuracy rate decreases by 2.3% compared to the original network. Compared to pruning and quantization, the compression ratio increases by 26.79% and 2.94%, respectively. The accuracy rate decreases by 0.9% and 2.3%, respectively. The source number estimation network proposed in this paper was well compressed, and with a small decrease in accuracy, this lightweight network still has a high accuracy rate compared to traditional methods and other convolutional neural networks.

**Table 5.** Comparison of source number estimation network compression results.

| | Model Size/MB | Average Accuracy | Compression Ratio | Declining Accuracy |
|---|---|---|---|---|
| Original Network | 12.92 | 96.7% | - | - |
| Pruning | 7.24 | 95.3% | 43.96% | 1.4% |
| Quantification | 4.16 | 95.8% | 67.80% | 0.9% |
| Pruning + Quantification | 3.78 | 94.4% | 70.74% | 2.3% |
| Other Networks | 2.47 | 86.4% | - | - |

Figure 9 shows the confusion matrix of recognition accuracy of LSNEN under different signal-to-noise ratios. When SNR = −5 dB, the classification accuracy can reach 92.0%. When SNR = 5 dB, the classification accuracy can reach 94.1%. When SNR = 15 dB, the classification accuracy can reach 97.5%. It can be seen that when the number of signals is one, the classification accuracy is higher.

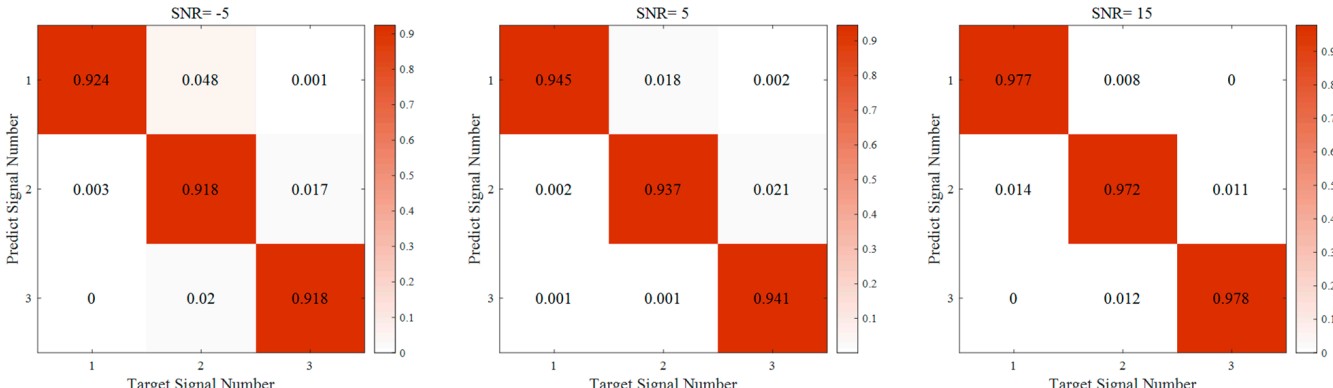

**Figure 9.** Confusion matrix on the simulation test set.

## 5. Conclusions

In this paper, a mixed signal source number estimation method based on deep learning is studied. The method of weight pruning and parameter quantization is combined to compress the information source number estimation network. This makes a lightweight source number estimation network more beneficial to be deployed on resource-limited devices. According to the complex convolution process, the convolution units in CNN are replaced with three-dimensional convolution units. The problem that CNN network architecture cannot fit complex mapping is solved. A network compression method combining weight pruning and quantization awareness is proposed to address the problem that the proposed network is difficult to deploy on mobile due to its large number of parameters and computational volume. The redundant connections in the network are removed, and the weights and activation values are quantized. As a result of our compression technique, the size of our network was reduced from 12.92 MB to 3.78 MB, resulting in a compression rate of 70.74%. Despite the reduction in size, the accuracy of source number identification in our experiments reached 94.4%. Undoubtedly, there are limitations to our work. Due to the constraints of experimental conditions, we have not been able to verify the proposed method on more comprehensive datasets. In future research, knowledge distillation is a promising method that could be explored to improve the model compression ratio and optimize the accuracy of the network.

**Author Contributions:** Conceptualization, W.S. and W.W.; methodology, W.S.; software, S.W.; validation, J.Z., S.W. and H.Z.; formal analysis, W.W.; investigation, W.S.; resources, W.S.; data curation, J.Z.; writing—original draft preparation, W.W.; writing—review and editing, W.S.; visualization, S.W. All authors have read and agreed to the published version of the manuscript.

**Funding:** This research received no external funding.

**Data Availability Statement:** Not applicable.

**Conflicts of Interest:** The authors declare no conflict of interest.

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
