# Peer review of "Pruning- and Quantization-Based Compression Algorithm for Number of Mixed Signals Identification Network"

_electronics, doi:10.3390/electronics12071694_

Round 1
Reviewer 1 Report
This paper introduced a lightweight source number estimation network (LSNEN) which can achieve robust estimation of the number of mixed complex signals at low SNR. Compared with other estimation methods, which require manual feature extraction, this network can realize the extraction of the depth feature of the original signal data. The convolutional neural network realizes complex mapping of modulated signals through the cascade of multiple three-dimensional convolutional modules. By using three-dimensional convolution module, the mapping of complex signal convolution is realized. The authors further proposed a compression method for the network to deploy the network in the mobile terminal with limited resources. This concept and investigation are interesting and important to the field, this study is data-rich, the figures are clean, and the manuscript is easy to follow. Thus, I recommended the manuscript for publication.
Some minor comments are listed below:
1. How to ensure and verify the estimation of the number of mixed complex signals are robust?
2. I guess SNR (signal-to-noise ratio) is also worth to have full name explanation in the Abstract :)
3. Why choose to use three-dimensional convolution module instead of other dimensions?
4. The authors mentioned the compression rate reached 70.74%. How is this number compared with the state-of-art results?
5. How scalable is the proposed approach?
6. What is the computational cost of the proposed approach?
7. What are the limitations of this work in practical applications? Please clarify.
8. The nomenclature should be included to help the reader to follow the paper conveniently.
Author Response
Dear reviewer,
Please cheak the attached respond letter.

Reviewer 2 Report
Dear authors,
Please see attached my review report.

Author Response

(The authors gave the same response as above.)

Reviewer 3 Report
In this paper, the authors proposed a pruning and quantization-based compression algorithm for a number of mixed signals identification networks.
The paper’s scope is within the scope of the journal, and it presents an original contribution. The abstract is sufficient to give useful information about the paper’s topic. The proposed algorithm is somehow described and thoroughly illustrated. The paper is well-structured and written, and the text is somehow clear and easy to read. However, there are some comments we recommend the authors to do:
In general, the abstract need to be rewritten in a better and more organized way. Also, at the end of the abstract, it is worthwhile to mention how much your proposed method is better than other estimation methods in terms of accuracy as percentages.
At the end of the introduction section, it is worthwhile to present a small paragraph about the organization or structure of the paper.
In the introduction section or where appropriate, you need to cite the following recent references about deep learning and CNNs in general:
Zhao, M.; Li, M.; Peng, S.-L.; Li, J. A Novel Deep Learning Model Compression Algorithm. Electronics 2022, 11, 1066. https://doi.org/10.3390/electronics11071066
Abuqaddom, I.; Mahafzah, B.; Faris, H. Oriented stochastic loss descent algorithm to train very deep multi-layer neural networks without vanishing gradients. Knowledge-Based Systems 2021, 230, 107391. https://doi.org/10.1016/j.knosys.2021.107391
Ghimire, D.; Kil, D.; Kim, S.-h. A Survey on Efficient Convolutional Neural Networks and Hardware Acceleration. Electronics 2022, 11, 945. https://doi.org/10.3390/electronics11060945
In the paper there are a lot of symbols, you may need to add two tables, one table in Section 2 and another one in Section 3, where these tables contain these symbols and a brief description/definition of them.
In Section 3, it is worthwhile to present the proposed algorithm as pseudocode and accordingly explain it briefly.
In Section 3, you need to mention whether your approach suffers from vanishing gradients or not and explain why, where you can cite the following reference regarding this issue:
Oriented stochastic loss descent algorithm to train very deep multi-layer neural networks without vanishing gradients.
In Section 4, you need to mention the experimental setup from hardware and software. Also, you may need to define and present some of the performance metrics as formulas, such as accuracy.
Also, in Section 4, the obtained results in Figures 7–8 and Tables 2 and 3 need more explanations and justifications. That is, you need to explain these results according to the proposed algorithmic design point of view.
At the end of the conclusions section, present your best results in comparison to other methods in terms of various performance metrics as percentages.
Author Response

(The authors gave the same response as above.)
